# DETECTING ADVERSARIAL EXAMPLES IS (NEARLY) AS HARD AS CLASSIFYING THEM

## ABSTRACT

Making classifiers robust to adversarial examples is challenging. Thus, many defenses tackle the seemingly easier task of *detecting* perturbed inputs.

We show a barrier towards this goal. We prove a general *hardness reduction* between detection and classification of adversarial examples: given a robust detector for attacks at distance $\epsilon$ (in some metric), we show how to build a similarly robust (but inefficient) *classifier* for attacks at distance $\epsilon/2$—and vice-versa.

Our reduction is computationally inefficient, and thus cannot be used to build practical classifiers. Instead, it is a useful sanity check to test whether empirical detection results imply something much stronger than the authors presumably anticipated.

To illustrate, we revisit $14$ empirical detector defenses published over the past years. For $12/14$ defenses, we show that the claimed detection results imply an inefficient classifier with robustness far beyond the state-of-the-art— thus casting some doubts on the results' validity.

Finally, we show that our reduction applies in both directions: a robust classifier for attacks at distance $\epsilon/2$ implies an inefficient robust detector at distance $\epsilon$. Thus, we argue that robust classification and robust detection should be regarded as (near)-equivalent problems, if we disregard their *computational* complexity.

## 1 INTRODUCTION

Building models that are robust to adversarial examples (Szegedy et al., 2014; Biggio et al., 2013) is a major challenge and open-problem in machine learning. Due to the inherent difficulty in building robust *classifiers*, researchers have attempted to build techniques to at least *detect* adversarial examples, a weaker task that is largely considered easier than robust classification (Xu et al., 2018; Pang et al., 2021; Sheikholeslami et al., 2021).

Yet, evaluating the robustness of empirical detector defenses is challenging. This is in part due to a lack of strong evaluation guidelines and benchmarks—akin to those developed for robust classifiers (Carlini et al., 2019; Croce et al., 2020)—as well as to a lack of long-standing comparative baselines such as adversarial training (Madry et al., 2018).

To illustrate, consider the following (fictitious) claims about two defenses against adversarial examples on CIFAR-10:

- defense A is a classifier that achieves robust accuracy of $90\%$ under $\ell_\infty$-perturbations bounded by $\epsilon = {}^4/255$;
- defense B also has a "rejection" option, and achieves robust accuracy of $90\%$ under $\ell_\infty$-perturbations bounded by $\epsilon = {}^8/255$ (we say that defense B is robust for some example if it classifies that example correctly, and either rejects/detects or correctly classifies all perturbed examples at distance $\epsilon$.)

*Which of these two (empirical) claims are you more likely to believe to be correct?*

Defense A claims much higher robustness than the current best result achieved with adversarial training (Madry et al., 2018; Rebuffi et al., 2021), the only empirical defense against adversarial examples that has stood the test of time. Indeed, the state-of-the-art $\ell_\infty$ robustness for $\epsilon = {}^4/255$

on CIFAR-10 (without external data) is $\approx 79\%$ (Rebuffi et al., 2021). Thus, the claim of defense A would likely be met with some initial skepticism and heightened scrutiny, as could be expected for such a claimed breakthrough result.

The claim of defense B is harder to assess, due to a lack of long-standing baselines for robust detectors (many detection defenses have been shown to be broken (Carlini & Wagner, 2017; Tramèr et al., 2020)). On one hand, detection of adversarial examples has largely been considered to be an easier task than classification (Xu et al., 2018; Pang et al., 2021; Sheikholeslami et al., 2021). On the other hand, defense B claims robustness to perturbations that are twice as large as defense A ($\epsilon = {}^{8}/_{255}$ vs. $\epsilon = {}^{4}/_{255}$).

*In this paper, we show that the claims of defenses A and B are, in fact, equivalent! (up to computational efficiency.)*

We prove a general *hardness reduction* between classification and detection of adversarial examples. Given a detector defense that achieves robust risk $\alpha$ for attacks at distance $\epsilon$ (under any metric), we show how to build an *explicit but inefficient* classifier that achieves robust risk $\alpha$ for classifying attacks at distance $\epsilon/2$. The reverse implication also holds: a classifier robust at distance $\epsilon/2$ implies an explicit but inefficient robust detector at distance $\epsilon$.

To the authors knowledge, there is no known way of leveraging computational *inefficiency* to build more robust models. We should thus be as "surprised" by the claim made by defense B as by the claim made by defense A.

Our reduction provides a way of assessing the plausibility of new robust detection claims, by contrasting them with results from the more mature literature on robust classification. To illustrate, we revisit 14 published detection defenses across three datasets, and show that in 12/14 cases the defense's robust detection claims would imply an inefficient classifier with robustness far superior to the current state-of-the-art. Yet, none of these detection papers make the claim that their techniques should imply such a breakthrough in robust *classification*.

Using our reduction, it is obvious that many detection defenses are claiming much stronger robustness than we believe feasible with current techniques. And indeed, many of these defenses were later shown to have overestimated their robustness (Carlini & Wagner, 2017; Tramèr et al., 2020).

Remarkably, we find that for *certified* defenses, the state-of-the-art results for provable robust classification and detection perfectly match the results implied by our reduction. For example, Sheikholeslami et al. (2021) recently proposed a certified detector on CIFAR-10 with provable robust error that is within $3\%$ of the provable error of the *inefficient* detector obtained by combining our result with the state-of-the-art robust classifier of Zhang et al. (2020a).

In summary, we prove that giving classifiers access to a detection option does not help robustness (or at least, not much). Our work provides, to our knowledge, the first example of a hardness reduction between different approaches for robust machine learning. As in the case of computational complexity, we believe that such reductions can be useful for identifying research questions or areas that are unlikely to bear fruit (bar a significant breakthrough)—so that the majority of the community's efforts can be redirected elsewhere.

On a technical level, our reduction exposes a natural connection between robustness and error correcting codes, which may be of independent interest.

## 2 Hardness Reductions Between Robust Classifiers and Detectors

In this section, we prove our main result: a reduction between robust detectors and robust classifiers, and vice-versa. We first introduce some useful notation and define the (robust) risk of classifiers with and without a detection option.

### 2.1 Preliminaries

We consider a classification task with a distribution $\mathcal{D}$ over examples $x \in \mathbb{R}^d$ with labels $y \in [C]$. A classifier is a function $f : \mathbb{R}^d \to [C]$. A detector is a classifier with an extra "rejection"

or "detection" option $\perp$, that indicates the absence of a classification. We assume for simplicity that classifiers and detectors are deterministic. Our results can easily be extended to randomized functions as well. The binary indicator function $\mathbb{1}_{\{A\}}$ is 1 if and only if the predicate $A$ is true.

We first define a classifier's *risk*, i.e., its classification error on unperturbed samples.

**Definition 1** (Risk). *Let $f : \mathbb{R}^d \to [C] \cup \{\perp\}$ be a classifier (optionally with a detection output $\perp$). The risk of $f$ is the expected rate at which $f$ fails to correctly classify a sample:*

$$R(f) := \mathbb{E}_{(x,y) \sim \mathcal{D}} \left[ \mathbb{1}_{\{f(x) \neq y\}} \right] \tag{1}$$

Note that for a detector, rejecting an unperturbed example sampled from the distribution $\mathcal{D}$ is counted as an error.

For classifiers without a rejection option, we define the *robust risk* as the risk on worst-case adversarial examples (Madry et al., 2018). Given an input $x$ sampled from $\mathcal{D}$, an adversarial example $\hat{x}$ is constrained to being within distance $d(x, \hat{x}) \leq \epsilon$ from $x$, where $d$ is some distance measure.

**Definition 2** (Robust risk). *Let $f : \mathbb{R}^d \to [C]$ be a classifier. The robust risk at distance $\epsilon$ is:*

$$R_{adv}^{\epsilon}(f) := \mathbb{E}_{(x,y) \sim \mathcal{D}} \left[ \max_{d(x,\hat{x}) \leq \epsilon} \mathbb{1}_{\{f(\hat{x}) \neq y\}} \right] \tag{2}$$

Thus, a sample $(x, y)$ is robustly classified if and only if every point within distance $\epsilon$ of $x$ (including $x$ itself) is correctly classified as $y$.

For a detector (a classifier with an extra detection/rejection output), we analogously define the robust risk with detection. The classifier is now allowed to reject adversarial examples.

**Definition 3** (Robust risk with detection). *Let $f : \mathbb{R}^d \to [C] \cup \{\perp\}$ be a classifier with an extra detection output $\perp$. The robust risk with detection at distance $\epsilon$ is:*

$$R_{adv\text{-}det}^{\epsilon}(f) := \mathbb{E}_{(x,y) \sim \mathcal{D}} \left[ \max_{d(x,\hat{x}) \leq \epsilon} \mathbb{1}_{\{f(x) \neq y \ \vee \ f(\hat{x}) \notin \{y, \perp\}\}} \right] \tag{3}$$

That is, a detector defense $f$ is robust on a natural input $x$ if and only if the defense classifies the natural input $x$ correctly, and the defense either rejects or correctly classifies every perturbed input $\hat{x}$ within distance $\epsilon$ from $x$. The requirement that the defense correctly classify natural examples eliminates pathological defenses that reject all inputs.

## 2.2 Robust Detection Implies Inefficient Robust Classification

We are now ready to introduce our main result, a reduction from a robust detector for adversarial examples at distance $\epsilon$, to an inefficient robust classifier at distance $\epsilon/2$. We later prove that this reduction also holds in the reverse direction, thereby demonstrating the equivalence between robust detection and classification—up to computational hardness.

**Theorem 4** ($\epsilon$-robust detection implies inefficient $\epsilon/2$-robust classification). *Let $d(\cdot, \cdot)$ be an arbitrary metric. Let $f$ be a detector that achieves risk $R(f) = \alpha$, and robust risk with detection $R_{adv\text{-}det}^{\epsilon}(f) = \beta$. Then, we can construct an explicit (but inefficient) classifier $g$ that achieves risk $R(g) \leq \alpha$ and robust risk $R_{adv}^{\epsilon/2}(g) \leq \beta$.*

*The classifier $g$ is constructed as follows on input $x$:*

- *Run the detector model $y \leftarrow f(x)$. If the input is not rejected, i.e., $y \neq \perp$, then output the label $y$ that was predicted by the detector.*

- *Otherwise, find an input $x'$ within distance $\epsilon/2$ of $x$ that is not rejected, i.e., $d(x, x') \leq \epsilon/2$ and $f(x') \neq \perp$. If such an input $x'$ exists, output the label $y \leftarrow f(x')$. Else, output a uniformly random label $y \in [C]$.*

An intuitive illustration for our construction, and for the proof of the theorem (see below) is in Figure 1.

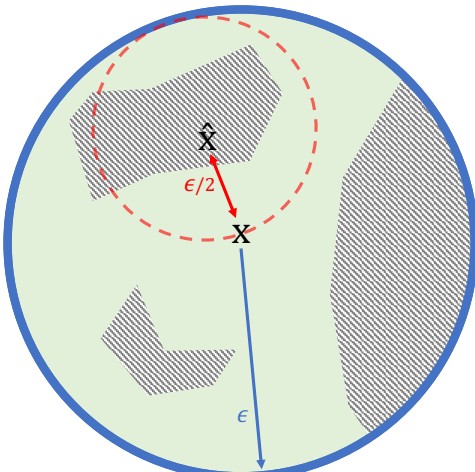

Figure 1: Illustration of the construction of a robust classifier from a robust detector in Theorem 4. The outer blue circle represents all inputs at distance at most $\epsilon$ from the input $x$. For a detector $f$, the areas in green correspond to correctly classified inputs, and ratcheted gray areas correspond to rejected inputs. The detector $f$ is thus robust on $x$ up to distance $\epsilon$. The classifier $g$ classifies a perturbed input $\hat{x}$, at distance $\epsilon/2$ from $x$, by finding any input within distance $\epsilon/2$ from $\hat{x}$ (the red dashed circle) that is not rejected by $f$. Such an input necessarily exists and is correctly labeled by $f$. The classifier $g$ is thus robust on $x$ up to distance $\epsilon/2$.

Our construction can be viewed as an analog of *minimum distance decoding* in coding theory. We can view a clean data point sampled from $\mathcal{D}$ as a codeword, and an adversarial example $\hat{x}$ as a noisy message with a certain number of errors (where the error magnitude is measured using an arbitrary metric on $\mathbb{R}^d$ rather than the Hamming distance that is typically used for error correcting codes). A standard result in coding theory states that if a code can *detect* $\alpha$ errors, then it can *correct* $\alpha/2$ errors. This result follows from a "ball-packing" argument: if $\alpha$ errors can be detected, then any two valid codewords must be at least at distance $\alpha$ from each other, and therefore $\alpha/2$ errors can be corrected via minimum distance decoding.

*Proof of Theorem 4.* First, note that the natural accuracy of our constructed classifier $g$ is at least as high as that of the detector $f$, since $g$ always mimics the output of $f$ whenever $f$ does not reject an input sampled from $\mathcal{D}$. Thus, $R(g) \leq R(f) = \alpha$.

Now, for the sake of contradiction, consider an input $(x, y) \sim \mathcal{D}$ for which the constructed classifier $g$ is not robust at distance $\epsilon/2$. By construction, this means that there exists some input $\hat{x}$ at distance $\epsilon/2$ from $x$ such that $\hat{x}$ is misclassified, i.e., $g(\hat{x}) = \hat{y} \neq y$. We will show that the detector $f$ is not robust with detection for $x$ either (for attacks at distance up to $\epsilon$).

By definition of the classifier $g$, if $g(\hat{x}) = \hat{y} \neq y$ then either:

- The detector $f$ also misclassifies $\hat{x}$, i.e., $f(\hat{x}) = \hat{y}$.

  So $f$ is not robust with detection for $x$ at distance $\epsilon$.

- There exists an input $x'$ within distance $\epsilon/2$ of $x$, such that the detector $f$ misclassifies $x'$, i.e. $f(x') = \hat{y}$.

  Note that by the triangular inequality, $d(x, x') \leq d(x, \hat{x}) + d(\hat{x}, x') \leq \epsilon/2 + \epsilon/2 = \epsilon$, and thus $f$ is not robust with detection for $x$ at distance $\epsilon$.

- The detector $f$ rejects all inputs $x'$ within distance $\epsilon/2$ of $x$ (and thus $g$ has output $\hat{y}$ by sampling a label at random).

  Since $d(x, \hat{x}) \leq \epsilon/2$, this implies that the detector also rejects the clean input $x$, i.e., $f(x) = \bot$, and thus $f$ is not robust with detection for $x$.

In summary, whenever the constructed classifier $g$ fails to robustly classify an input $x$ up to distance $\epsilon/2$, the detector $f$ also fails to robustly classify $x$ with detection up to distance $\epsilon$. Taking expectations over the entire distribution $\mathcal{D}$ concludes the proof. □

Note that the classifier $g$ constructed in Theorem 4 is computationally inefficient. Indeed, the second step of the defense consists in finding a non-rejected input within some metric ball. If the original detector $f$ is a non-convex function (e.g., a deep neural network), then this step consists in solving an intractable non-convex optimization problem. Our reduction is thus typically not suitable for building a practical robust classifier. Instead, it demonstrates the existence of an inefficient but *explicit* robust classifier. We discuss the implications of this result more thoroughly in Section 3.

A corollary to our reduction is that many "information theoretic" results about robust classifiers can be directly extended to robust detectors. For example, Tsipras et al. (2019) prove that there exists a formal tradeoff between a classifier's clean accuracy and robust accuracy for certain natural tasks. Since their result applies to *any* classifier (including inefficient ones), combining their result with our reduction implies that a similar accuracy-robustness tradeoff exists for detectors. More precisely, Tsipras et al. (2019) show that for certain classification tasks and suitable choices of parameters $\alpha, \beta, \epsilon$, any classifier $g$ which achieves risk $R(g) \leq \alpha$ must have robust risk at least $R_{\text{adv}}^{\epsilon}(g) \geq \beta$ against $\ell_\infty$-perturbations bounded by $\epsilon$. By our reduction, this implies that any detector $f$ with risk at most $R(f) \leq \alpha$ must also have robust risk with detection at least $R_{\text{adv-det}}^{\epsilon/2}(f) \geq \beta$ against $\ell_\infty$-perturbations bounded by $\epsilon/2$.

Similar arguments can be applied to show, for instance, that the increased data complexity of robust generalization from Schmidt et al. (2018), or the tradeoff between robustness to multiple perturbation types from Tramèr & Boneh (2019), also apply to robust detectors.

Our reduction does not apply for "computational" hardness results that have been shown for robust classification. For example, Garg et al. (2020) and Bubeck et al. (2018) show ("unnatural") distributions where learning a robust classifier is computationally hard—under standard cryptographic assumptions. We cannot use Theorem 4 to conclude that learning a robust *detector* is hard for these distributions, since the existence of such a detector would only imply an inefficient robust classifier which does not contradict the results of Garg et al. (2020) or Bubeck et al. (2018).

## 2.3 ROBUST CLASSIFICATION IMPLIES INEFFICIENT ROBUST DETECTION

A similar argument as in Theorem 4 can be used in the opposite direction, to show that a robust classifier at distance $\epsilon/2$ implies an inefficient robust detector at distance $\epsilon$. Taken together, Theorem 4 and Theorem 5 show that robust detection and classification are *equivalent*, up to a factor 2 in the norm bound and up to computational constraints.

**Theorem 5** ($\epsilon/2$ robust-classification implies inefficient $\epsilon$-robust detection). *Let $d(\cdot, \cdot)$ be an arbitrary metric. Let $g$ be a defense that achieves robust risk $R_{adv}^{\epsilon/2}(f) = \beta$. Then, we can construct an explicit (but inefficient) defense $f$ that achieves risk $R(f) \leq \beta$ and robust risk with detection $R_{adv\text{-}det}^{\epsilon}(f) \leq \beta$.*

*The defense $f$ is constructed as follows on input $x$:*

- *Run the classifier $y \leftarrow g(x)$.*

- *Find a perturbed input $x'$ withing distance $\epsilon/2$ of $x$ that is classified differently, i.e., $d(x, x') \leq \epsilon/2$ and $g(x') \neq y$. If such an input $x'$ exists, reject the input and output $\perp$. Else, output the class $y$.*

We provide the proof of Theorem 5 in Appendix A.

A main distinction between Theorem 4 and Theorem 5 is that the construction in Theorem 4 preserves clean accuracy, but the construction in Theorem 5 does not. That is, the constructed robust detector in Theorem 5 has *clean* accuracy that is equal to the robust classifier's *robust* accuracy.

The construction in Theorem 5 can be efficiently (but approximately) instantiated by a *certifiably robust* classifier (Wong & Kolter, 2018; Raghunathan et al., 2018). These defenses can certify that a classifier's output is constant for all points within some distance $\epsilon$ of the input. For an adversarial

example $\hat{x}$ for $g$, the certification always fails and thus the constructed detector $f$ will reject $\hat{x}$. If $g$ is robust and the certification succeeds, the detector $f$ copies the output of $g$. However, a certified defense may fail to certify a robust input (a false negative), and thus the detector $f$ may reject more inputs than with the "optimal" construction in Theorem 5. This reduction from a certified classifier to a detector is implicit in (Wong & Kolter, 2018, Section 3.1).

## 3   WHAT ARE DETECTION DEFENSES CLAIMING?

We now survey 14 detection defenses, and consider the robust *classification* performance that these defenses implicitly claim (via Theorem 4). As we will see, in 12/14 cases, the defenses' detection results imply an inefficient classifier with far better robust accuracy than the state-of-the-art.

Before presenting our experimental setup and the explicit results from the reduction, we first discuss how we believe these results should be interpreted.

**Interpreting our reduction.**   Suppose that some detector defense claims a robust accuracy that implies—via our reduction—an inefficient classifier with much higher robustness that the state-of-the-art (e.g., the defense A described in the introduction of this paper).

A first possible interpretation of our reduction is that this robust detector implies the *existence* of a robust classifier. This interpretation is rather weak however, since it is typically presumed that robust classification is possible, and that human perception is one concrete example of a robust classifier. The mere existence of a robust classifier is thus typically already assumed to be true.

Our reduction yields a stronger result. It provides an *explicit construction* of an (inefficient) robust classifier from a robust detector. The question then is whether we should expect the construction of inefficient robust classifiers to be easier than the construction of efficient ones. That is, do we expect that we can leverage computational inefficiency to build more robust classifiers that the current state-of-the-art?

We do not know of a positive answer to this question, and there is evidence to suggest that the answer may be negative.[1] For example, the work of Schmidt et al. (2018) proves that for a synthetic classification task between Gaussian distributions, building more robust classifiers requires additional *data* regardless of the amount of computation power. Their results are corroborated by current state-of-the-art robust classifiers based on adversarial training (Madry et al., 2018), which do not appear to be limited by computational constraints. On CIFAR-10 for example, adversarial training achieves 100% robust *training* accuracy (Schmidt et al., 2018). Thus, it is unclear how computational inefficiency could be leveraged to build more robust classifiers using existing techniques.

Candidate approaches could be to train much larger models (e.g., with an exponential number of parameters), or to perform an exhaustive architecture search to find more robust models. Yet, note that the robust classifier constructed in our reduction only uses its unbounded computational power at *inference time*. That is, the classifier that is built in Theorem 4 uses a trained detector model as a subroutine (which is presumed to be efficient), and then solves a non-convex optimization problem at inference time. The classifier built in our reduction is thus presumably weaker than a robust classifier that can be *trained* with unbounded computational power.

To summarize, when a detector defense claims a certain robust accuracy, this implies the existence of a *concretely instantiatable robust classifier with an inefficient inference procedure*. If this inefficient classifier is much more robust than the current state-of-the-art, this does not necessarily mean that the defense's claims is *wrong*. But given how challenging robust classification is proving to be, we have reason to be skeptical of such a major breakthrough (even for inefficient classifiers). To compound this, many proposed detection defenses are quite *simple*, and reject adversarial inputs based on some standard statistical test over a neural network's features. It would thus be particularly surprising if such simple techniques could yield robust *classifiers*, given that "simple" approaches to adversarial robustness (denoising, compression, randomness, etc.) are ineffective (He et al., 2017).

---

[1] Some works have shown that for certain "unnatural distributions", computational inefficiency is necessary to build robust classifiers (Garg et al., 2020; Bubeck et al., 2018). Yet, since we presume that the human perceptual system *is* robust to small perturbations on natural data (e.g., such as CIFAR-10), there must exist some efficient natural process to achieve robustness on such data.

As a result, it is not too surprising that a number of the detector defenses that we survey have already been broken by stronger attacks (Carlini & Wagner, 2017; Tramèr et al., 2020). Our reduction would have already suggested that such a break was likely to happen.

**Experimental setup.** We choose 14 detector defenses from the literature (see Table 1). Our selection of these defenses was partially motivated by a pragmatic consideration on the easiness of translating the defenses' claims into a bound on the robust risk with detection $R^\epsilon_{\text{adv-det}}$. Indeed, some defenses simply report a single AUC score for the detector's performance, from which we cannot derive a useful bound on the robust risk. We thus focus on defenses that either directly report a robust error akin to Definition 3, or that provide concrete pairs of false-positive and false-negative rates (e.g., a full ROC curve). In the latter case, we compute a "best-effort" bound on the robust risk with detection[2] as:

$$R^\epsilon_{\text{adv-det}}(f) \leq \text{FPR} + \text{FNR} + R(f) \,, \tag{4}$$

where FPR and FNR are the detector's false-positive and false-negative rates for a fixed detection threshold, and $R(f)$ is the defense's standard risk (i.e., the test error on natural examples).

The above union bound in Equation (4) is quite pessimistic, as we may over-count examples that lead to multiple sources of errors (e.g., a natural input that is misclassified and erroneously detected). The true robustness claim made by these detector defenses might thus be stronger than what we obtain from our bound. We encourage future detection papers to report their adversarial risk with detection, $R^\epsilon_{\text{adv-det}}$, to facilitate direct comparisons with robust classifiers using our reduction.

The 14 detector defenses use three datasets: MNIST, CIFAR-10 and ImageNet, and consider adversarial examples under the $\ell_\infty$ or $\ell_2$ norms. Given a claim of robust detection at distance $\epsilon$, we contrast it to a state-of-the-art robust classification result for distance $\epsilon/2$:

- On MNIST with $\ell_\infty$ attacks, we use the adversarially-trained TRADES classifier (Zhang et al., 2019) and measure robust error with the Square attack (Andriushchenko et al., 2020).

- On MNIST with $\ell_2$ attacks, we use the adversarially-trained classifier from Tramèr & Boneh (2019) and measure robust error with PGD (Madry et al., 2018).

- On CIFAR-10, for both $\ell_\infty$ and $\ell_2$ attacks we use the adversarially-trained classifier of Rebuffi et al. (2021) (trained without external data), and attack it using the APGD-CE attack from AutoAttack (Croce & Hein, 2020).

- For ImageNet, for both $\ell_\infty$ and $\ell_2$ attacks we use adversarially-trained classifiers and PGD attacks from Engstrom et al. (2019).

We also consider two *certified* defenses for $\ell_\infty$ attacks on CIFAR-10: the robust classifier of Zhang et al. (2020a), and a recent certified detector of Sheikholeslami et al. (2021).

**Results.** As we can see from Table 1, most defenses claim a detection performance that implies a far greater robust accuracy than our current best robust classifiers. To illustrate with a concrete example, the CIFAR-10 detector of Miller et al. (2019) claims to achieve robust accuracy with detection of 75% for $\ell_2$ attacks with $\epsilon = 2.9$. Using Theorem 4, this implies an inefficient classifier with robust accuracy of 75% for $\ell_2$ attacks with $\epsilon = {}^{2.9}/2 = 1.45$. Yet, the current state-of-the-art robust accuracy for such a perturbation budget is only 30% (Rebuffi et al., 2021). If this detector defense's robustness claim were correct, it would imply a remarkable breakthrough in robust classification.

Why do many of these defenses claim robust accuracies that appear "too good to be true"? A primary reason is that the vast majority of the above detector defenses do not consider evaluations against *adaptive attacks* (Carlini et al., 2019; Athalye et al., 2018; Tramèr et al., 2020). That is, these defenses show that they can detect *some fixed attacks*, and thereafter conclude that the detector is robust against *all attacks*. As in the case of robust classifiers, such an evaluation is clearly insufficient! Some defenses do evaluate against adaptive adversaries, but fail to build a sufficiently strong attack to reliably approximate the worst-case robust risk. Because of the lack of a strong comparative baseline, it is not always immediately clear that these results are overly strong.

---

[2]Many detector defenses report performance against a set of *fixed* (non-adaptive) attacks. We interpret these results as being an approximation of the worst-case risk.

Table 1: For each detector defense, we compute a (best-effort) bound on the claimed robust risk with detection $R_{\text{adv-det}}^{\epsilon}$ using Equation (4), and report the complement (the robust accuracy with detection), $1 - R_{\text{adv-det}}^{\epsilon}$. For each detector's robustness claim (at distance $\epsilon$), we report the state-of-the-art robust classification accuracy for attacks at distance $\epsilon/2$, denoted $1 - R_{\text{adv}}^{\epsilon/2}$. Detection defense claims that imply a higher robust classification accuracy than the current state-of-the-art are highlighted in red.

| Dataset | Defense | Norm | $\epsilon$ | $1 - R_{\text{adv-det}}^{\epsilon}$ | $1 - R_{\text{adv}}^{\epsilon/2}$ |
|---------|---------|------|------------|------------------------------------|----------------------------------|
| MNIST | Grosse et al. (2017) | $\ell_\infty$ | 0.5 | $\geq 98\%$ | 94% |
| | Ma et al. (2018) | $\ell_2$ | 4.2 | $\geq 99\%$ | 72% |
| | Raghuram et al. (2021) | $\ell_2$ | 8.9 | $\geq 74\%$ | 0% |
| CIFAR-10 | Yin et al. (2020) | $\ell_2$ | 1.7 | $\geq 90\%$ | 66% |
| | Feinman et al. (2017) | $\ell_2$ | 2.7 | $\geq 43\%$ | 36% |
| | Miller et al. (2019) | $\ell_2$ | 2.9 | $\geq 75\%$ | 30% |
| | Raghuram et al. (2021) | $\ell_2$ | 4.0 | $\geq 56\%$ | 10% |
| | Ma & Liu (2019) | $\ell_\infty$ | $^4/_{255}$ | $\geq 96\%$ | 85% |
| | Roth et al. (2019) | $\ell_\infty$ | $^8/_{255}$ | $\geq 66\%$ | 79% |
| | Lee et al. (2018) | $\ell_\infty$ | $^{20}/_{255}$ | $\geq 81\%$ | 59% |
| | Li et al. (2019) | $\ell_\infty$ | $^{26}/_{255}$ | $\geq 80\%$ | 44% |
| ImageNet | Xu et al. (2018) | $\ell_2$ | 1.0 | $\geq 67\%$ | 54% |
| | Ma & Liu (2019) | $\ell_\infty$ | $^2/_{255}$ | $\geq 68\%$ | 55% |
| | Jha et al. (2019) | $\ell_\infty$ | $^2/_{255}$ | $\geq 30\%$ | 55% |
| | Hendrycks & Gimpel (2017) | $\ell_\infty$ | $^{10}/_{255}$ | $\geq 76\%$ | 30% |
| | Yu et al. (2019) | $\ell_\infty$ | $^{26}/_{255}$ | $\geq 7\%$ | 5% |

For example, the recent work of Raghuram et al. (2021, ICML Long Talk) builds a detector on MNIST with a FNR of $\leq 5\%$ at a FPR of $\leq 20\%$, for *adaptive* $\ell_2$ attacks bounded by $\epsilon = 8.9$. Yet, this perturbation bound is much larger than the average distance between an MNIST image and the nearest image from a different class! Thus, an attack within this perturbation bound can trivially reduce the detector's accuracy to chance. On CIFAR-10, the same detector achieves $95\%$ clean accuracy, and a FNR of $\leq 19\%$ at a FPR of $\leq 20\%$ for *adaptive* $\ell_2$ attacks bounded by $\epsilon = 4$. Using Equation (4), this yields a bound on the robust accuracy with detection of $1 - R_{\text{adv-det}}^{\epsilon}(f) \geq 1 - (5\% + 19\% + 20\%) = 56\%$. In contrast, the best robust classifier we are aware of for $\ell_2$ attacks bounded by $\epsilon = 2$ achieves robust accuracy of only $10\%$ (Rebuffi et al., 2021). In summary, the adaptive attack considered in this detector defense's evaluation is highly unlikely to be good approximation of a worst-case attack, and this defense can likely be broken by stronger attacks.

**Certifiably robust detection and classification.** In Table 2, we look at the robust accuracy with detection, and standard robust accuracy achieved by *certified* defenses (for which the claimed robustness numbers are necessarily mathematically correct).

We note that our reduction is not as meaningful in the case of certified defenses, since it is highly plausible that computational inefficiency *can* be leveraged to build better certified classifiers. Indeed, given any robust classifier (e.g., an adversarially trained model), the classifier's robustness can always be certified inefficiently (by enumerating over all points within an $\epsilon$-ball). Thus, the existence of an inefficient classifier with higher certified robustness than the state-of-the-art is to be expected.

Nevertheless, we find that existing results for certified classifiers and detectors perfectly match what is implied by our reduction (up to $\pm 2\%$ error). For example, Zhang et al. (2020a) follow a long line of results on robust classifiers and achieve $39\%$ robust accuracy on CIFAR-10 for perturbations of $\ell_\infty$-norm below $^4/_{255}$. Together with Theorem 5, this implies an inefficient detector with $39\%$ robust

Table 2: Certified robust accuracy $1 - R_{\text{adv}}^{\epsilon/2}$ for the defense of Zhang et al. (2020a), and certified robust accuracy with detection $1 - R_{\text{adv-det}}^{\epsilon}$ for the defense of Sheikholeslami et al. (2021).

| $\epsilon$ | $1 - R_{\text{adv-det}}^{\epsilon}$ | $1 - R_{\text{adv}}^{\epsilon/2}$ |
|---|---|---|
| $^8/_{255}$ | 37% | 39% |
| $^{16}/_{255}$ | 32% | 33% |

detection accuracy for perturbations of $\ell_\infty$-norm below $^8/_{255}$. The recent work of Sheikholeslami et al. (2021) nearly matches that bound (37% robust accuracy with detection), with a defense that has the advantage of being concretely efficient.

These results give additional credence to our thesis: with current techniques, robust classification is indeed approximately twice as hard (in terms of the perturbation bounds covered) than robust detection.

**Extensions and open problems.** The main open problem raised by our work is of course whether it could be possible to show an *efficient* reduction between classification and detection of adversarial examples, but this seems implausible (at least with our minimum distance decoding approach).

Another interesting question is whether a similar reduction can be shown for robustness to less "structured" perturbations than $\ell_p$ balls and other metric spaces. For example, there has been a line of research on defending against *adversarial patches* (Brown et al., 2017), using empirical (Hayes, 2018; Naseer et al., 2019; Chou et al., 2020) and certifiable techniques (Chiang et al., 2020; Zhang et al., 2020b; Xiang et al., 2021). To use our result, we would have to define some metric to measure the size of an adversarial patch's perturbation. Yet, the size of a patch is typically defined by the number of contiguously perturbed pixels, which does not define a metric (in particular, it does not satisfy the triangular inequality which our reduction relies on).

Finally, similar hardness reductions might exist between other candidate approaches for building robust classifiers. For example, the question of whether (test-time) randomness can be leveraged to build more robust models is also intriguing .Empirical defenses that use randomness can be notoriously hard to evaluate (Athalye et al., 2018; Tramèr et al., 2020), so a reduction similar to ours might be useful in showing that we should not expect such approaches to bare fruit.

## 4 CONCLUSION

We have shown formal reductions between robust classification with, and without, a detection option. Our results show that significant progress on one of these two tasks implies similar progress on the other—unless computational inefficiency can somehow be leveraged to build more robust models. This raises the question on whether we should spend our efforts on studying both of these tasks, or focus our efforts on a single one.

On one hand, the two tasks represent different ways of tackling a common goal, and working on either task might result in new techniques or ideas that apply to the other task as well. On the other hand, our reductions show that unless we make progress on both tasks, work on one of the tasks can merely aim to match the robustness of our inefficient constructions, whilst improving their computational complexity.

We believe our reduction will serve as a useful sanity-check when assessing the claims of future detector defenses. Detector defenses' robustness evaluations have received less stringent scrutiny than robust classifiers over the past years, perhaps in part due to a lack of strong comparative baselines. Instead of having to wait until some detector defense's claims pass the test-of-time, we show that detection results can be directly contrasted against long-standing results for robust classification.

When applying this approach to past detector defenses, we find that many make robustness claims that imply significant breakthroughs in robust classification. We believe our reduction could have been useful in highlighting the suspiciously strong claims made by many of these defenses—before they were explicitly broken by stronger attacks.

**Ethics statement.** Our paper demonstrates a fundamental barrier towards detecting against adversarial examples, under the assumption that our current techniques are insufficient to achieve strong (inefficient) robust classification. We do not however explicitly break any existing defenses (our results merely strongly suggest that many existing detector defenses' claims are suspiciously high). Our paper therefore cannot lead to any explicit harms, but aims to further our understanding of the hardness of robust classification and detection.

**Reproducibility statement.** Our paper's contribution is mainly of theoretical nature. Section 2 is self-contained and clearly states our assumptions, results and proofs (except for the proof of Theorem 5 in Appendix A). The experiments in Section 3 use only public datasets and pre-trained models, with clearly indicates hyper-parameters for all attacks that we evaluate.

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

## A    PROOF OF THEOREM 5.

We recall Theorem 5:

**Theorem 5** ($\epsilon/2$ robust-classification implies inefficient $\epsilon$-robust detection). *Let $d(\cdot, \cdot)$ be an arbitrary metric. Let $g$ be a defense that achieves robust risk $R_{adv}^{\epsilon/2}(f) = \beta$. Then, we can construct an explicit (but inefficient) defense $f$ that achieves risk $R(f) \le \beta$ and robust risk with detection $R_{adv\text{-}det}^{\epsilon}(f) \le \beta$.*

*The defense $f$ is constructed as follows on input $x$:*

- *Run the classifier $y \leftarrow g(x)$.*

- *Find a perturbed input $x'$ withing distance $\epsilon/2$ of $x$ that is classified differently, i.e., $d(x, x') \le \epsilon/2$ and $g(x') \ne y$. If such an input $x'$ exists, reject the input and output $\bot$. Else, output the class $y$.*

*Proof of Theorem 5.* Note that for any input $(x, y)$ for which the classifier $g$ is robust at distance $\epsilon/2$, no input $x'$ above exists and so $f(x) = y$. Thus, the risk of $f$ is at most the robust risk of $g$, so $R(f) \le \beta$.

Now, consider an input $(x, y) \sim \mathcal{D}$ for which $f$ is not robust with detection at distance $\epsilon$. That is, either $f(x) \ne y$, or there exists an input $\hat{x}$ at distance $d(x, \hat{x}) \le \epsilon$ such that $f(\hat{x}) = \hat{y} \notin \{y, \bot\}$. We will show that the defense $g$ is not robust for $x$ either (for attacks at distance up to $\epsilon/2$.)

If $f(x) \ne y$, then by the same argument as above it cannot be the case that $g$ is robust at distance $\epsilon/2$ for $x$.

So let us consider the case where $f(\hat{x}) = \hat{y} \notin \{y, \bot\}$. By the definition of $f$, this means that for all $x'$ at distance at most $\epsilon/2$ from $\hat{x}$, we have $g(x') = \hat{y}$. But, note that there exists a point $x^*$ that is at distance at most $\epsilon/2$ from both $\hat{x}$ and $x$. Since we must have $g(x^*) = \hat{y}$, we conclude that $g$ is not robust at distance $\epsilon/2$ for $x$.

Taking expectations over the distribution $\mathcal{D}$ concludes the proof. □

