# OpenReview forum: "Detecting Adversarial Examples Is (Nearly) As Hard As Classifying Them"
_ICLR.cc/2022/Conference — ICLR 2022 Submitted_

### Official Review · Reviewer_2ebX · 2021-10-25

**Correctness:** 3
**Technical Novelty And Significance:** 3
**Empirical Novelty And Significance:** 3
**Recommendation:** 5
**Confidence:** 4

**Main Review:**

Strength:
This work connects the areas of adversarial training and adversarial detection and the writing is easy to follow. It is of great significance to the future development of adversarial detection research.

Weakness：
The construction of equivalent classifiers and detectors in Theorem 4 and 5 are interesting. However, I have concerns about theorem 4 that epsilon-robust detection implies inefficient epsilon/2-robust classification. For one thing, it lacks a practical and efficient solution to find a perturbed input that is classified differently; for another, the claim is not necessary in practice, for example, if the input is rejected by an adversarial detection, practitioners can simply add random noise on the input until the new input is not rejected. And by the way, compared to the analysis of robust detection of robust classifying, considering adaptive attacks towards adversarial classification and adversarial detection simultaneously is more important.

The title of this submission seems inappropriate. It’s widely acknowledged the success rate of adversarial detection far outweighs the success rate of adversarial classifying in previous studies. To be specific, since a large amount of high dimensional images within an epsilon ball, classifying all images is a difficult task for neural networks with limited capacity. By contrast, the detection task just has to tell the difference between adversarial examples and natural examples. Hence, the findings in this submission are not convincing that detecting adversarial examples is as hard as classifying them. Similar difficulties should be indicated by similar success rates, otherwise, it will give the public misunderstanding.

A modest proposal: avoid using the paragraphing abstract and delete '\emph{}' and '---' in the open review submission page.


**Summary Of The Paper:**

This submission connects the areas of adversarial training and adversarial robust detection. Generally speaking, I think the conclusions in the submitted paper can provide beneficial insights for the community, and avoid overclaims in future adversarial detection research.

**Summary Of The Review:**

In general, I think this submission can provide beneficial insights and evaluations for the field of adversarial detection. However, it lacks practical solutions, which implies limited contributions. I hope my concerns about Theorem 4 and 5 can be considered (or pointed out my misunderstanding) in the discussion stage. Though I cannot recommend acceptance at this stage, I will increase my score if my concerns are solved properly.

---

> ### Author Response · Authors · 2021-11-12
> **Response**
>
> We thank the reviewer for their comments and questions.
>
> We do not fully understand some of the points raised by the reviewer, and we would thus be grateful if the reviewer could clarify these points (see below) to enable a continued discussion.
>
> > *“For one thing, it lacks a practical and efficient solution to find a perturbed input that is classified differently”*
>
> Yes, we acknowledge this repeatedly and candidly in the paper. Our result is only information theoretic. We discuss at length why we believe that this result is interesting despite this limitation. Instantiating our construction efficiently is not at all a goal of our paper.
>
> > *“the claim is not necessary in practice, for example, if the input is rejected by an adversarial detection, practitioners can simply add random noise on the input until the new input is not rejected”*
>
> It is not clear to us what the reviewer means by this. Clearly, if a detector can be bypassed simply by adding random noise to the input, then that detector isn't robust. Our result only applies to detectors that are actually robust.
>
> > *“And by the way, compared to the analysis of robust detection of robust classifying, considering adaptive attacks towards adversarial classification and adversarial detection simultaneously is more important”*
>
> Could the reviewer please clarify what they mean by this?
>
> > *“The title of this submission seems inappropriate. It’s widely acknowledged the success rate of adversarial detection far outweighs the success rate of adversarial classifying in previous studies. To be specific, since a large amount of high dimensional images within an epsilon ball, classifying all images is a difficult task for neural networks with limited capacity. By contrast, the detection task just has to tell the difference between adversarial examples and natural examples. Hence, the findings in this submission are not convincing that detecting adversarial examples is as hard as classifying them.”*
>
> Our paper **proves** that robust detection of adversarial examples is as hard as robust classification. The title of our paper states exactly this. In what way is this inappropriate? The whole point of our paper is that the prevailing wisdom that detection is easier than classification is, in fact, wrong! The reviewer may intuitively believe that detection is easier than classification in high dimensions. But we explicitly **prove** otherwise. It is not clear to us how we can make this result more “convincing” than by stating and proving a formal theorem.
>
> > “Similar difficulties should be indicated by similar success rates, otherwise, it will give the public misunderstanding.”
>
> Could the reviewer please clarify what they mean by this?

---

> > ### Author Response · Authors · 2021-11-19
> > **Remaining questions and request for clarifications**
> >
> > Since we are approaching the end of the discussion period, we would like to enquire if the reviewer has any remaining questions or further points to discuss after our response.
> > Moreover, there were a number of points raised in the review that we did not understand (see our original response), and we would appreciate if the reviewer could clarify these.

---

> > ### Comment · Reviewer_2ebX · 2021-11-20
> > **Response**
> >
> > Thanks for your response. There are some clarifications to help the author better understand my comments.
> >
> > Adversarial examples are generated by adding specific noise to the natural examples. Note it is specific noise. For a robust model, by adding random noise (e.g. Gaussian noise) to natural data, the generated examples are unable to mislead the model prediction. In the scenario of this submission, ‘practitioners can simply add random noise on the input until the new input is not rejected’ seems a simple and practical solution
> >
> > I agree with the conclusion by Reviewer Jyt1 that ‘what you show is that the maximum eps for which eps-robust classifier h exist is equal to max eps for which 2-eps-robust-with-rejection classifiers exist.’  In my view, the title seems inappropriate.
> >
> > For a fixed maximum eps in the same dataset (e.g., CIFAR-10), the detection success rate of adversarial detection is far beyond the classification success rate by adversarial training. The title ‘Detecting Adversarial Examples Is (Nearly) As Hard As Classifying Them’ may give researchers a misunderstanding that adversarial detection (by detection methods) and adversarial classifying (e.g., by adversarial training) have the similar success rate.

---

> > > ### Author Response · Authors · 2021-11-20
> > > **Response**
> > >
> > > > Adversarial examples are generated by adding specific noise to the natural examples. Note it is specific noise. For a robust model, by adding random noise (e.g. Gaussian noise) to natural data, the generated examples are unable to mislead the model prediction. In the scenario of this submission, ‘practitioners can simply add random noise on the input until the new input is not rejected’ seems a simple and practical solution
> > >
> > > Whether random noise can mislead a robust model or not depends on the amount of noise added. If the noise is sufficiently large, random noise will definitely mislead a model, even if the model is robust.
> > > It seems that you're suggesting that one could add random noise to a rejected example, until the example is no longer rejected, and then return the predicted class. But there's absolutely no guarantee that the amount of random noise that is needed to make a change in the detector's output will not also change the predicted class.
> > >
> > > > I agree with the conclusion by Reviewer Jyt1 that ‘what you show is that the maximum eps for which eps-robust classifier h exist is equal to max eps for which 2-eps-robust-with-rejection classifiers exist.’ In my view, the title seems inappropriate.
> > > For a fixed maximum eps in the same dataset (e.g., CIFAR-10), the detection success rate of adversarial detection is far beyond the classification success rate by adversarial training. The title ‘Detecting Adversarial Examples Is (Nearly) As Hard As Classifying Them’ may give researchers a misunderstanding that adversarial detection (by detection methods) and adversarial classifying (e.g., by adversarial training) have the similar success rate.
> > >
> > > We also agree with the interpretation of Reviewer Jyt1. This is why in our paper, we compare the robustness of detectors at distance $\epsilon$, to classifiers at distance $\epsilon/2$.
> > >
> > > Our reduction states that these two regimes are equal (up to computational hardness).
> > >
> > > This is why we say "nearly as hard" in our title. An alternative title could have been "Detecting Adversarial Examples Is At Most Half As Hard As Classifying Them". But when speaking of hardness of problems (e.g., in computational complexity), it is typical to omit constant factors. That is why we say that robust detection is (nearly) as hard as classification, because there's at most a factor 2 that separates them.

---

### Official Review · Reviewer_RAPn · 2021-10-31

**Correctness:** 2
**Technical Novelty And Significance:** 2
**Empirical Novelty And Significance:** 1
**Recommendation:** 3
**Confidence:** 4

**Main Review:**

While the idea is interesting and needs some attention; however, considering only the attacker is intelligent is somewhat wrong or misleading in both this and existing papers referred. If the defense can be broken using a new adaptive attack using the knowledge of the defense, then the defender must also have the same knowledge to modify the defense to carefully check the limit of defense.

Another interesting question is why the attacker needs to generate an attack if he/she can assume all knowledge of the defense and the target model? Why not simply modify the decision? Therefore, before showcasing the defenses are not working it is expected to consider this and adaptive defense or give the same freedom to the defender as well.

The claims made in the paper are not empirically tested, they are mainly based on the assumptions of the authors or one selected paper. It will be great if the authors can showcase that actually, the defenses are not working.

What do the authors mean by the " the evaluation is inefficient" on page 7? Please refer to some string defenses:

[1] Detection based defense against adversarial examples from the steganalysis point of view. In Proceedings of the IEEE/CVF Conference on Computer Vision and Pattern Recognition (pp. 4825-4834).
[2] DAMAD: Database, Attack, and Model Agnostic Adversarial Perturbation Detector, In IEEE Transactions on Neural Networks and Learning Systems (TNNLS), 2021.
[3] Image Transformation based Defense Against Adversarial Perturbation on Deep Learning Models, In IEEE Transactions on Dependable and Secure Computing (TDSC), 2021, vol. 18, no. 5, pp. 2106-2121.

Similarly again the term "worst-case robust attack" is not clear? It seems like the paper is based on some pre-assumptions that the detector can not work. Please clarify?

Next sentence on page 7: " if this detector defense's robustness claims were correct -- ". not clean what the author meant and downgrade take the credit of the defense work?

Most of the assumptions are based on one work only: Rebuffi et al. 2021. I feel the authors need to revisit the paper and literature thoroughly not only a few papers which showcase that existing defenses do not work or depend on the robsutness paper.


**Summary Of The Paper:**

The paper showcases the vulnerability of existing adversarial detection algorithms. Multiple existing detection-based defense algorithms are considered and have shown theoretically that the robustness claimed by those algorithms might not be the same as claimed in the corresponding papers.

**Summary Of The Review:**

The paper is based on assumptions and findings of one single paper most of the time. Without explicit showcase that the existing defense will not work, the analysis seems misleading and unfair to the parts of the defenders as well.

The authors believe that the existing defenses do not resemble the "worst-case attack". Please generate such an attack and showcase without explicitly touching the defenses that defense is not working.

In my knowledge the true and fair concept to both attacker and defender needs to be there to make serious progress in the field, else this can be just another paper reflecting the singularities of the defenses.

---

> ### Author Response · Authors · 2021-11-12
> **Response**
>
> We thank the reviewer for their comments.
>
> Many of the reviewer’s criticisms seem to have nothing to do with our paper specifically, but rather with the entire literature on adversarial examples. If this is indeed the case, we encourage the reviewer to explicitly state this and to disentangle their criticisms of our specific paper from their criticisms of the field that our paper builds on (criticisms of the field can of course be valuable and we try to clarify some important points below).
>
> > *Summary Of The Paper: ...*
>
> This is not an accurate summary of our paper.
> First and foremost, our paper proves a **reduction** between robust classification and detection. This is the main contribution of our paper, and isn’t even mentioned in the above summary!
> To demonstrate an application of this reduction, we use it to convert the claims made by robust detectors into claims of robust classification, for which we have a better sense of the current state-of-the-art.
> Under this view, we find that existing detectors, **if correct**, would imply a huge breakthrough in robust classification.
>
> > *“considering only the attacker is intelligent is somewhat wrong or misleading in both this and existing papers referred. If the defense can be broken using a new adaptive attack using the knowledge of the defense, then the defender must also have the same knowledge to modify the defense”*
>
> This comment conflates “intelligent” (a subjective property) and “adaptive” (a well-defined game-theoretic property).
> The defender can be very intelligent and design a very smart defense, but ultimately this defense has to be deployed, after which an adversary can adapt their attack strategy. Adaptive evaluations of defenses have a long history in security and cryptography and those fields  universally agree that this is the right way of evaluating a defense.
>
> > *“Another interesting question is why the attacker needs to generate an attack if he/she can assume all knowledge of the defense and the target model? Why not simply modify the decision?”*
>
> This comment conflates “knowledge” and “control”. An attacker can have “knowledge” of how a model works, without the ability to arbitrarily “control” the model’s inputs and outputs.
>
> > *“What do the authors mean by the "the evaluation is inefficient"*
>
> We don't say this. The correct quote is "the evaluation is **insufficient**".
> This means the evaluated attack fails to find adversarial examples when they exist.
>
> > *“the term "worst-case robust attack" is not clear”*
>
> We don't say this.  We use the term "**worst-case attack**", which is a standard term to designate the strongest possible attack (a.k.a. the worst-case from the perspective of the defender). This is an attack that always finds an adversarial example if one exists.
>
> > *“Please refer to some strong defenses:*
>
> Paper [1] claims detection accuracy above 98% on ImageNet for l-infinity attacks bounded by eps=8/255. By our theorem, this would imply a huge breakthrough in robust ImageNet classification if correct.
> Again, this does not mean that [1] is necessarily incorrect, but its claims should naturally be met with high scrutiny, given our belief that a huge breakthrough in robust classification is very hard to achieve.
>
> Paper [2] and Paper [3] consider attacks that are **unbounded** (e.g, I-FGSM with epsilon=1). A detector defense cannot possibly be robust against unbounded perturbations, so it is obvious that these papers were not evaluated against strong attacks.
>
> > *“It seems like the paper is based on some pre-assumptions that the detector can not work.”*
>
> No, we don't pre-assume that the detector cannot work.
> We show that **IF** the detector did work as claimed, this would **imply** a very surprising result.
>
> > *“Most of the assumptions are based on one work only: Rebuffi et al. 2021”*
>
> Rebuffi et al. is the last in a long line of work on adversarial training that has been very thoroughly evaluated.
> It is not clear what "assumptions" we are basing on this work? We simply use this work as the current SOTA for adversarially robust classifiers (which the community agrees that it is, see e.g., https://robustbench.github.io)
>
> > *“The authors believe that the existing defenses do not resemble the "worst-case attack". Please generate such an attack and showcase that defense is not working”*
>
> The question of whether these defenses are actually correct or not is orthogonal to our main point.
> We prove that **IF** these defenses work as claimed, then they've also made a huge breakthrough in robust classification. This is clearly a surprising implication since none of these detector defenses makes this claim (if you had written a paper that achieves some very strong new result, wouldn’t you want to take credit for it?)
>
> Our work is thus interesting because it contextualizes these detectors’ claims in light of the more mature literature on robust classification (e.g., thanks to rigorous community benchmarks like robustbench mentioned above).

---

> > ### Author Response · Authors · 2021-11-19
> > **Remaining questions?**
> >
> > Since we are approaching the end of the discussion period, we would like to enquire if the reviewer has any remaining questions or further points to discuss after our response.
> > In particular, we hope that our response managed to clarify the message and contributions of our paper.

---

> > > ### Comment · Reviewer_RAPn · 2021-11-19
> > > **Questions**
> > >
> > > Thanks for your response.
> > >
> > > Can you please clarify:
> > >
> > > We show that IF the detector did work as claimed, this would imply a very surprising result.
> > >
> > > AND
> > >
> > > We prove that IF these defenses work as claimed, then they've also made a huge breakthrough in robust classification.
> > >
> > > What is meant by so-called breakthrough results according to the authors or algorithm?
> > >
> > > Are these papers (or many more of these types) have breakthrough results?
> > >
> > > NIC: Detecting adversarial samples with neural network invariant checking,” in Proc. Netw. Distrib. Syst. Secur. Symp., 2019.
> > > DAMAD: Database, Attack, and Model Agnostic Adversarial Perturbation Detector, In IEEE Transactions on Neural Networks and Learning Systems (TNNLS), 2021.
> > >
> > > What is missing to make the evaluation "sufficient"? How someone will find the future attacks to fool the current defense?

---

> > > > ### Author Response · Authors · 2021-11-20
> > > > **Response**
> > > >
> > > > > Can you please clarify:
> > > > We show that IF the detector did work as claimed, this would imply a very surprising result.
> > > > AND
> > > > We prove that IF these defenses work as claimed, then they've also made a huge breakthrough in robust classification.
> > > > What is meant by so-called breakthrough results according to the authors or algorithm?
> > > >
> > > > The breakthrough result here is a robust *classifier* with robustness far beyond the current state-of-the-art.
> > > > Since it is generally believed that building more robust classifiers is quite challenging, a paper that claims (even indirectly) to have made huge progress on this problem would be a big breakthrough if true.
> > > >
> > > > We can offer the following analogy, from our response to reviewer hr6H: suppose that some new paper claims to have proved some theorem A. The theorem’s proof is quite complicated so it is hard for the community to evaluate its correctness. Then someone else shows a very simple and easily checkable proof that theorem A, if true, would imply that P≠NP. This does necessarily mean that the original paper’s proof of theorem A is incorrect. But it should rightfully make people a lot more suspicious of the first paper, and the authors of that paper may have to argue why their techniques should suddenly solve such a longstanding hard problem.
> > > >
> > > > The problem we’re concerned with here isn’t quite as fancy as P≠NP, but the same intuition applies. A paper that claims a very robust (even inefficient) classifier would be a big result that would have to be evaluated with great care. And so any paper that proposes a very robust detector should rightfully receive the same scrutiny, since our paper shows that these are actually equivalent.
> > > >
> > > > > Are these papers (or many more of these types) have breakthrough results?
> > > >
> > > > - NIC is one of the papers that we evaluate in Table 1. This paper claims at least 68% robust accuracy with detection for perturbations of size 2/255 on ImageNet. By our reduction, this implies an inefficient classifier with robust accuracy of 68% for perturbations of size 1/255. This robust accuracy is ~30% higher than the current state-of-the-art, so this would indeed be a very surprising and breakthrough result, if true.
> > > > - As we noted in our first response, DAMAD claims robust detection of *unbounded* perturbations. This is impossible, since an unbounded perturbation can always transform a clean sample from one class into a clean sample from another class. It is thus impossible to be accurate and robust for unbounded attacks.
> > > > - There are many more detection defenses that we did not study in this paper. Our goal is not to consider every defense, this would be impossible. Rather, we show that a large number of detector defenses in the literature actually imply very strong robust classification results.
> > > >
> > > > > What is missing to make the evaluation "sufficient"? How someone will find the future attacks to fool the current defense?
> > > >
> > > > Designing good evaluations and building strong attacks are challenging problems. Carlini et al. (https://arxiv.org/abs/1902.06705) and Tramer et al. (https://arxiv.org/abs/2002.08347) give some guidelines, and strong attack benchmarks like robustbench give a good baseline. But many of these guidelines and benchmarks have been proposed with classifiers in mind, rather than detectors.
> > > > That's why properly evaluating a robust detector remains quite challenging today.
> > > >
> > > > Our paper offers an alternative, which is to convert detector claims into equivalent classifier claims, which are easier to compare to the existing state-of-the-art.

---

### Official Review · Reviewer_Jyt1 · 2021-11-02

**Correctness:** 4
**Technical Novelty And Significance:** 3
**Empirical Novelty And Significance:** 3
**Recommendation:** 6
**Confidence:** 4

**Main Review:**

Please list both the strengths and weaknesses of the paper. When discussing weaknesses, please provide concrete, actionable feedback on the paper.

On the positive side: the connection between detection and classification is a very natural question that deserves attention. The paper proves a simple, but very nice theorem that as far as I know was not proved before. The connection between “testing” and “decoding” is not completely new in coding theory, and the paper also mentions this, but observing this phenomenon in the context of robust learning, as far as I know, is new. Also, I like the fact that the paper uses this connection to study the implications of results already claimed in the literature.

On the down side, the fact that the theorem of the paper is proved using information theoretic (rather than computationally efficient) reductions, limits the ways one can benefit from such a connection. In particular:
1. It is not completely fair to question previous works’ claimed results on robustness using detection, because those papers did not claim “certified results” or “information theoretic” results, but rather “hardness” of breaking their schemes. Since the reductions in this paper are not computationally feasible, one cannot conclude that those schemes were indeed not secure *computationally*.
2. Most, if not all, settings in which robust learning is a hot topic, one already knows the existence of a “robust” ground truck function. In particular, for image classification, it is the assumption that humans are robust to small \eps perturbations, and the goal is to find such classifiers automatically. In such contexts, the results of this paper become obsolete, because the reductions exist trivially. This further limits the applicability of the results.

Other comments:

Page 2: “To the author's knowledge, there is no known way of leveraging computational inefficiency to build more robust models. ”

The (cited) paper by Garg et al seems to exactly show the possibility that computational efficiency could be leveraged to achieve robustness.

The two papers “Garg et al. (2020) and Bubeck et al. (2018) are actually quite different, in how they deal with the role of computational efficiency” One deals with poly-time learning and the other deals with poly-time attacking. I think it's the latter that is more directly related to this work’s message.

Following up on the issue (2) mentioned above, one might say that the result of the paper still applies even if the ground truth is *not* robust to eps perturbations. But then in that case, it brings up another issue: the definition used in this paper would *not* imply that an *adversarial example* is actually misclassified. This issue is discussed in some previous work such as “Revisiting Adversarial Risk” Suggala et al (AISTA’19) and “Adversarial Risk and Robustness: General Definitions...” Diochnos et al (NeurIPS’18).


**Summary Of The Paper:**

Adversarial examples are test time attacks in which the input is modified by up to distance \eps (under some metric) and the goal of adversarially robust learning is to have high (generalized) accuracy even under such attacks. One way to make predictions is to always output a label. Another way is to “abstain/detect” when the learner thinks the input is not clean and is perturbed with. The way we evaluate the performance in the detection model is to count detected perturbed inputs as “correctly classified”.

The paper asks a very natural question: is it easier to learn when detection/abstain is allowed or not? The main result of the paper is very clean: For any metric d, and any eps, the existence of a learner that achieves accuracy c under detection model under 2\eps perturbations is (information theoretically) equivalent to the existence of a classifier in the no-detection model with accuracy c and eps perturbation.

The proof is “constructive” but it is not “efficiently constructive”. Namely, given a classifier in either of the two settings above, the paper shows a rather simple (but smart) way of constructing another classifier (with the parameters stated above) in the other model.

The paper then takes this connection to revisit the results of quite a few papers from the literature in which they have claimed defenses that use detection as their key idea. The paper observes that the bounds that (many) of those papers claim would imply classifiers with no detection/abstain that beat the state of the art adversarially robust classifiers. The paper cautiously claims that this indicates that the defenses of those papers are not actually secure, but rather “not broken” under simple attacks tried by the authors.

**Summary Of The Review:**

The paper proves a natural theoretical result that is at the heart of robust learning. The result is information theoretic, but I still find it quite natural.

I think one should be very cautious to not overly interpret the implications of this paper, but I think the mere theoretical observation that testing and decoding in the context of adversarial learning are equivalent has a merit, that at least puts this paper on the border for ICLR.

---

> ### Author Response · Authors · 2021-11-12
> **Response**
>
> We thank the reviewer for their positive assessment of our paper, and for their insightful comments.
>
> > *“It is not completely fair to question previous works’ claimed results on robustness using detection, because those papers did not claim “certified results” or “information theoretic” results, but rather “hardness” of breaking their schemes”*
>
> We agree with this point, but we note that none of the detection papers we consider explicitly make this distinction. The threat model considered in these papers typically only places a bound on the perturbation size used by the adversary.
>
> > *“Most, if not all, settings in which robust learning is a hot topic, one already knows the existence of a “robust” ground truck function. In such contexts, the results of this paper become obsolete, because the reductions exist trivially.”*
>
> This is not quite true, and we discuss this subtlety in Section 3.
> A result that would merely imply the **existence** of a robust classifier would indeed be quite weak, since we assume that humans are such a robust classifier.
> But our result yields something conceptually stronger, namely an **explicit construction** (albeit an inefficient one).
> Even in settings like computer vision where we presume that robust classifiers exist, an explicit construction of such a classifier remains elusive and providing such a construction (even if inefficient) would be a very big breakthrough.
>
> > *“The (cited) paper by Garg et al seems to exactly show the possibility that computational efficiency could be leveraged to achieve robustness”*
>
> Yes, what we meant to say here is that “there is no known way of leveraging computational inefficiency to build more robust models **for natural tasks**”. The work of Garg et al. shows that robustness is hard for a specific artificial data distribution that is embedded with cryptographic objects.
>
> > *“one might say that the result of the paper still applies even if the ground truth is not robust to eps perturbations”*
>
> The settings we had in mind were ones where the ground truth is robust. If this is not the case, we agree that our definitions would have to be amended.

---

> > ### Comment · Reviewer_Jyt1 · 2021-11-20
> > **Re:**
> >
> > > but we note that none of the detection papers we consider explicitly make this distinction.
> >
> > This is a good point.
> >
> > > But our result yields something conceptually stronger, namely an explicit construction (albeit an inefficient one).
> >
> > I am not still sure if I agree completely with this exact phrasing. If we assume that eps-robust classifier h exist, then getting them from 2-eps-robust classifier h' with rejection is trivially true. Moreover, your mapping of h' to h is not done in poly time. In fact, even if we assume that a poly-time eps-robust h exists, you don't find one.
> >
> > Having said that, I think what you can say is that your reductions have a quantitative aspect. Namely, what you show is that the maximum eps for which eps-robust classifier h exist is equal to max eps for which 2-eps-robust-with-rejection classifiers exist. This, as I said in my review, is a basic and interesting result, and hence I maintain my positive score.

---

> > > ### Author Response · Authors · 2021-11-20
> > > **Re**
> > >
> > > > If we assume that eps-robust classifier h exist, then getting them from 2-eps-robust classifier h' with rejection is trivially true
> > >
> > > We're not sure we understand this point. The existence of an eps-robust classifier is indeed trivially true, if we start by assuming such a classifier exists. But that doesn't mean that we're able to write down a concrete algorithm for such an eps-robust classifier (even for an inefficient algorithm).
> > >
> > > E.g., we may assume that humans are a robust classifier, but that doesn't bring us any closer to writing down a robust algorithm, which is what our reduction does.
> > > And if someone were to propose such a concrete algorithm, even if inefficient, this would probably be viewed as an amazing result (even if we already assumed the existence of such an algorithm).
> > >
> > > Does this distinction make sense?

---

> > > > ### Comment · Reviewer_Jyt1 · 2021-11-22
> > > > **Re:**
> > > >
> > > > Yes. what you say makes sense. I think it would be very helpful if you formally elaborate more on the constructively of your approach. Namely, as I understand, you go from one type of classifier (with or without rejection) to the other type without even calling the ground truth. (While if one could call the ground truth infinitely many times, we could reconstruct it.)

---

> > > > > ### Author Response · Authors · 2021-11-22
> > > > > **Computational complexity vs data complexity**
> > > > >
> > > > > Yes, that is a very good observation!
> > > > >
> > > > > In statistical learning theory, we usually care about two different types of complexity: computational complexity(how much time the algorithms take to run) and data complexity (how many samples from the distribution we need).
> > > > >
> > > > > Our reduction preserves the data complexity of the original detector.
> > > > > This is what makes the reduction nontrivial, as an algorithm with infinite queries could indeed just learn the exact distribution.
> > > > >
> > > > > We will clarify this in our write-up.

---

> > > > > > ### Comment · Reviewer_Jyt1 · 2021-11-22
> > > > > > **optimal Bayes classifiers**
> > > > > >
> > > > > > Regarding ground truth c being robust: as I said, your results actually seem to be applicable even if c is not robust, but with a careful choice of what it means to be an adversarial example. Let me explain that more:
> > > > > >
> > > > > > Suppose c is not actually eps robust under D and eps perturbations. Also, suppose $f$ is such that $\Pr_{(x,y) \approx D}[\exists x' \in Ball_\epsilon(x): f(x) \neq y] \leq \delta$. Then, you show how to find $f'$ with rejection, which achieves the same error $\delta$ under $2 \epsilon$ perturbations. Note that this is true, even if $c$ was not robust under $\epsilon$ perturbations of examples coming form $D$. The subtle catch is that, in this case, you automatically end up using what is usually referred to as the "corrupted input" definition of adversarial example here: https://proceedings.neurips.cc/paper/2018/file/3483e5ec0489e5c394b028ec4e81f3e1-Paper.pdf
> > > > > > which coincides with other definitions when $c$ is actually robust.
> > > > > >
> > > > > > Putting things together, as I understand, you are in fact proving that: using the corrupted-input definition of Adversarial Examples, even if the ground truth is not robust, the error of the Bayes optimal classifier for $\epsilon$-perturbations without rejection is equal to the error of the Bayes optimal classifier for $2\epsilon$ perturbations with rejection.

---

> > > > > > > ### Author Response · Authors · 2021-11-24
> > > > > > > **Response**
> > > > > > >
> > > > > > > Yes, that makes sense. Thanks for the clarification.
> > > > > > >
> > > > > > > Our reduction shows that if your detector is robust around a certain prediction, you can create a classifier that is robust around the same prediction. This holds regardless of whether that prediction is actually correct.
> > > > > > > So we can also apply our reduction to a setting where we define robustness with respect to the model's "clean" prediction.

---

### Official Review · Reviewer_hr6H · 2021-11-03

**Correctness:** 3
**Technical Novelty And Significance:** 4
**Empirical Novelty And Significance:** 2
**Recommendation:** 5
**Confidence:** 4

**Main Review:**

We summarize the Strengths and Weakness of this paper as following:

**Strengths**:
1. It is the first paper to attempt to unify the detection and robust classifier approaches.
2. The authors provide constructive steps to reduce the detector to classifier and vice versa.
3. The part about certified defense highlights the strictness of the theoretical results.

**Weakness**:
1. In the paper, the authors prove that one can (theoretically) construct a robust (inefficient) classifier from a robust detector which has equivalent robustness (Theorem 4). However, the constructive steps provided in Theorem 4 can not be exactly solved in practice for most of the defense models (except for certified defenses). If it is extremely hard to figure out such a classifier, we still could not say: there exists such a feasible classifier with the same robustness as the detection model. Therefore, the robust accuracy reported by the constructed classifier could be unreasonably high (similar evidence can be found in Table 1). As a result, the construction process in Theorem 4 might give a false sense about the true robustness of the classifier, as well as the original detector.
2. One possible direction is that: one could approximately solve the “inefficient” problem in Theorem 4, using gradient methods or black box optimization methods. Imagine that, one can approximately solve this problem, he/she can get such a classification model in practice, which will greatly improve the potential practical use of the proposed theorem.
3. Based on the discussion above, we also hesitate about the practical importance of the experimental results in Table 1. The correctness of the theorems needs verification of the irrelevance of computational complexity in reduction, which is not provided. And the arguments of suspicious high performance of detection defenses are based on adaptive attacks instead of their theorems. Therefore, as said in Section 3, Interpreting our reduction, “does not mean that the defense’s claims is wrong”, the experiment results shown in table 1, which is based on the theorems, can not be used to make any conclusion about the performance of existing detection defenses.
4. Since most of the results of this paper are based on norm-based attacks. Are the results valid for other types of attacks, such as spatial attacks and so on?

**Other comments or remarks**
A typo in theorem 4 proof: in the second and third bullets, x should be $\hat{x}$.



**Summary Of The Paper:**

This paper considers one important question: how to fairly compare the adversarial robustness between detection-based defenses and classification-based defenses. From the theoretical perspective, the authors show that: one can always (ideally) construct a robust classifier from a robust detector which has equivalent robustness, and vice versa. Based on this construction method, they are able to transfer the robustness between robust detectors and robust classifiers. Finally, they find that most existing detection defenses achieve suspiciously high robust performance compared with state-of-art robust classifiers, if they apply the proposed “transferring” criteria.

**Summary Of The Review:**

Overall the paper considers an interesting problem and tries to unify the detection and classifier defenses. However, the theorem is only correct without considering the computational complexity and poses another question about the relation between robust classifier training and computational complexity. Also, the experiment part should include approximate results of the reduction steps to verify the feasibility of the theorems in practice.

---

> ### Author Response · Authors · 2021-11-12
> **Response**
>
> We thank the reviewer for their insightful comments and questions.
>
> > *“As a result, the construction process in Theorem 4 might give a false sense about the true robustness of the classifier, as well as the original detector”*
>
> Yes, we acknowledge this fact and discuss it thoroughly in Section 3.
> What our theorem implies is that **IF** these detectors’ claims are correct, then they've actually solved a seemingly very hard problem, namely that of building an inefficient robust classifier. Since we don’t currently know how to build inefficient robust classifiers, these detectors’ claims imply a very surprising breakthrough---if they are indeed correct.
>
> Consider the following analogy: a new paper claims to have proved some theorem A. The theorem’s proof is quite complicated so it is hard for the community to evaluate its correctness. Then someone else shows a very simple and easily checkable proof that theorem A, if true, would imply that P≠NP. This does necessarily mean that the original paper’s proof of theorem A is incorrect. But it should rightfully make people a lot more suspicious of the first paper, and the authors of that paper may have to argue why their techniques should suddenly solve such a longstanding hard problem.
>
> The problem we’re concerned with here isn’t quite as fancy as P≠NP, but the same intuition applies. A paper that claims a very robust (even inefficient) classifier would be a big result that would have to be evaluated with great care. And so any paper that proposes a very robust detector should rightfully receive the same scrutiny.
>
>
> > *“One possible direction is that: one could approximately solve the “inefficient” problem in Theorem 4, using gradient methods or black box optimization methods”*
>
> This is not at all our aim in this work. Even if a good approximation would exist, it is not clear what one would do with such an approximation. Presumably one could try to convert a detector into an efficient classifier, and then evaluate the robustness of this classifier. But this wouldn’t imply anything meaningful about the original detector (i.e., even if the constructed efficient classifier turns out to not be robust, this could just be because the approximation was not good enough).
>
>
> > *“The correctness of the theorems needs verification of the irrelevance of computational complexity in reduction, which is not provided.”*
>
> Our theorems are **correct** (that is, unless the reviewer believes there is a mistake in the proofs). The theorems merely state that a robust detector would imply an inefficient robust classifier (and vice-versa).
> We do not know of any proof that building an inefficient robust classifier is as hard as building an efficient one. But we do discuss some existing evidence for this claim, such as the work of Schmidt et al. that shows that this statement is true in some settings.
>
>
> > *“the experiment results shown in table 1, which is based on the theorems, can not be used to make any conclusion about the performance of existing detection defenses”*
>
> No, we cannot make any definite claims (exactly like in the analogy above, we cannot guarantee that the paper’s proof of Theorem A, which implies P≠NP, is incorrect).
> But this doesn’t mean that our theorems aren’t useful as a sanity check! At the very least, we should expect papers that propose very robust detectors to provide some argument as to why the techniques they use should be expected to yield much more robust classifiers as well.
>
>
> > *“Are the results valid for other types of attacks, such as spatial attacks and so on”*
>
> Yes, as long as the perturbation bounds can be embedded with a meaningful distance measure. E.g., one could adapt our theorem to rotations and translations as well. But we are not aware of any detector defenses that evaluate against spatial attacks.

---

> > ### Author Response · Authors · 2021-11-19
> > **Remaining questions?**
> >
> > Since we are approaching the end of the discussion period, we would like to enquire if the reviewer has any remaining questions or further points to discuss after our response.

---

> > ### Comment · Reviewer_hr6H · 2021-11-24
> > **Reply to rebuttal.**
> >
> > We thank the author's answer to our questions. However, we still have the concerns.
> >
> > **Towards the analogy as the author mentioned: “a new paper claims to have proved some theorem A. The theorem’s proof is quite complicated so it is hard for the community to evaluate its correctness. Then someone else shows a very simple and easily checkable proof that theorem A, if true, would imply that P≠NP. ”**
> >
> > We agree with the logic in the analogy explained by the author. However, this paper fails to provide “a very simple and easily checkable proof for Theorem A”. In particular for this paper, it is entirely possible that the super high performance of the constructed inefficient robust classifier comes from the infinite power to determine the existence of non-rejected points in eps/2 ball. If the reduction were realistic (polynomial) and the accuracy of the constructed efficient robust classifier were still super high, we could then conclude the detection results were superficial. However, the only realistic results in the paper are certified case, and the results are consistent with the theorem, based on which we can not make any conclusion.
> >
> > **Even if a good approximation would exist, it is not clear what one would do with such an approximation.**
> >
> > The reason I gave the suggestion is to use empirical results to verify the practical usefulness of the theorem as a sanity check. For example, you can use the approximation of the conversion in certified defense comparison. If the results were consistent not only in analysis but also in practical reductions, at least this could give evidence to exclude the influence of the complexity in reduction.
> >
> > **Our theorems are correct (that is, unless the reviewer believes there is a mistake in the proofs)**
> >
> > Yes, I agree that the proof is correct.
> >
> > **Summary** This paper tries to unify the view of detection and classifier, which is interesting. But I am still concerned with the inefficient reduction which can not be ignored. Unless there is strong evidence to show the reduction is unrelated with the complexity, I will keep my score.

---

> > > ### Author Response · Authors · 2021-11-24
> > > **Computational complexity vs data complexity**
> > >
> > > We disagree that our reduction is not useful because it is not polynomial time.
> > >
> > > In machine learning, the data complexity of a learning algorithm is typically a lot more important than its computational complexity.
> > > (i.e., we know that we can learn any function given infinite *data*, but given a finite training set it is often not clear how to leverage infinite *compute* to learn better models).
> > >
> > > The same is true for robustness today. We do not know, a priori, of any way to leverage infinite compute to build more robust classifiers on natural datasets. (But there are many papers that show how to get better robustness if you collect more data).
> > >
> > > Because of this, even a reduction that is computationally inefficient can be interesting, if it preserves the data complexity (which our reduction does).
> > >
> > > Our empirical results can be interpreted as follows: if some detectors' claims are correct, then we have found a way to build more robust classifiers *without collecting additional data*. We believe that such an implication is interesting even if it is not constructive due to its computational inefficiency, as it sheds light onto the relative data complexities of robust classification and detection.
> > >
> > > We will clarify this point further in our paper.

---

### Decision · Program_Chairs · 2022-01-20

**Decision:**

Reject

**Comment:**

The paper investigates a very interesting problem of the connections between adversarial detection and adversarial classification. Theoretically, the authors show that one can always (ideally) construct a robust classifier from a robust detector that has equivalent robustness, and vice versa. This theorem is only correct without considering the computational complexity. However, the authors did not provide any approximate results of the reduction steps to verify the feasibility of the theorems in practice, which is the main concern of all reviewers. So we can say the paper is a reminder to the community we need to be careful about the detection results but did not provide any evidence to say they are overclaimed (only a conjecture based on the theorem in the paper) which greatly limits the contribution of the paper. Due to the competitiveness of ICLR, I cannot recommend accepting it.